# Swept Source Optical Coherence Tomography Analysis of the Selected Eye’s Anterior Segment Parameters

**DOI:** 10.3390/jcm10051094

**Published:** 2021-03-05

**Authors:** Michał Dembski, Anna Nowińska, Klaudia Ulfik-Dembska, Edward Wylęgała

**Affiliations:** 1Clinical Department of Ophthalmology, Faculty of Medical Sciences in Zabrze, Medical University of Silesia in Katowice, 40-055 Katowice, Poland; klaudia.ulfik@gmail.com (K.U.-D.); rekroz@sum.edu.pl (E.W.); 2Ophthalmology Department, Railway Hospital in Katowice, 40-760 Katowice, Poland; anna.nowinska@sum.edu.pl

**Keywords:** optical coherence tomography, anterior chamber angle, keratometry, cornea

## Abstract

Background: The present study determined the mean reference values of the anterior segment parameters of the selected eye using swept source optical coherence tomography (SS-OCT) in healthy Caucasian participants. Methods: A total of 166 volunteers (age 54–79 years), women (*n* = 92) and men (*n* = 74), were analyzed. One eye of each subject was randomly selected for anterior segment imaging. The anterior segment of the eye was scanned with CASIA2. The analyzed anterior segment parameters were divided into three groups, namely parameters of the cornea, lens, and angle. Results: The OCT (e.g., Ks, Kf, pKf, pKs, and central corneal thickness) and Fourier parameters of the cornea were significantly different between females and males. The iridocorneal angle was the smallest in the upper quadrant for all distance from the apex of the angle (250, 500, and 750 µm). Conclusions: Therefore, SS-OCT enables the analysis of parameters of the cornea, anterior chamber, lens, and iridocorneal angle, highlighting its clinical utility. Sex-specific differences in the analyzed parameters should be taken into account during the diagnosis of corneal diseases. The configuration of the filtration angle is an important marker during glaucoma diagnosis and drainage implant surgery. Measurements with CASIA 2is characterized by very good repeatability.

## 1. Introduction

Examination of the anterior segment of the eye is an element of basic ophthalmic diagnostics and in clinical practice it is used to diagnose and monitor the progression of many ophthalmic diseases. Modern imaging modalities of the anterior segment of the eye include ultrasonography, high-resolution magnetic resonance imaging, Scheimpflug imaging, and optical coherent tomography (OCT) [1,2,3,4]. Each technique allows for the morphological and morphometric analyses of the cornea, anterior chamber, filtration angle, and lens; however, due to significant differences in their methods of image analysis, the obtained morphometric results cannot be interchangeably used among the various techniques [5,6]. Ultrasonography is the least used method due to the low image resolution, difficulties in correcting distortions, long scanning time, and invasiveness of the examination [1]. Meanwhile, Scheimpflug imaging and OCT have the greatest potential due to the high scanning speed, contactless operation, and possibility of collecting a large amount of data from measurements of the anterior segment of the eye in vivo [2,3,4,5,6]. OCT is a precise, contactless technique for obtaining high-resolution tissue images. The first CT image of the retina was obtained in 1991, and the first commercially available OCT device was introduced in 1996 [7].Over the last 20 years, OCT has been widely applied for the diagnosis of various diseases of the anterior and posterior segments of the eye, as well as for intraoperative evaluations [8,9].Currently, several OCT techniques are available, which differ in terms of the resolution of the obtained images, duration of the examination, range of the analyzed parameters (posterior and anterior segments of the eye), and possibility of performing morphometric measurements. There are three main OCT techniques: time domain OCT, spectral or Fourier domain OCT (FD-OCT), and OCT using swept source lasers (swept source OCT, SS-OCT). The imaging method of all these OCT techniques is characterized by the use of a reference mirror with a longitudinal and transverse resolution of, respectively, 18 and 60 μm, which allows for obtaining images at a speed of 2048 A-scans/s. In 2002, the introduction of OCT combined with Fourier analysis of the light beam spectrum allowed for obtaining images with a longitudinal and transverse resolution of respectively 5 and 15 μm. The elimination of moving elements of the reference mirror significantly reduced the time required of image acquisition (26,000 A-scans/s) [10,11]. Subsequently, the next generation of OCT devices using swept source lasers was introduced in 2005 [12]. The commercially available device CASIA2 (Tomey Corporation, Nagoya, Japan) has a scanning speed of 50,000 A-scans/s, longitudinal resolution of 10·μm, and transverse resolution of 30·μm. In this generation of OCT devices, the optical coherence has also improved, allowing for obtaining images with a greater sensitivity and at a greater depth. SS-OCT is characterized by high reproducibility of the results of the anterior segment of the eye, as demonstrated in previous studies [13,14]. The scan frame size is 16 mm × 13 mm, which allows for obtaining data from the surface of the cornea to the posterior surface of the lens in a single scan. Moreover, the second-generation CASIA devices are the first to enable automatic measurement and calculation of the radius of anterior lens (RAL) with high repeatability and reproducibility regardless of the accommodative stress [15]. At present, however, the reference values for many parameters of the anterior segment of the eye, specifically the lens parameters and the iridocorneal angle, for different age groups remains unknown. Therefore, similar to that of the retina and optic nerve, a normative database of parameters of the anterior segment of the eye using SS-OCT must be established for the diagnosis of various pathologies [16]. Furthermore, this database may lay the foundation for further research including patients with diseases such as pseudoexfoliation syndrome, angle-closure glaucoma, dystrophies, or keratoconus.

## 2. Materials and Methods

The present study was performed at the Chair and Department of Ophthalmology, Faculty of Medical Sciences in Zabrze, Medical University of Silesia in Katowice, from October 2019 to March 2020. A total of 166 eyes (age, 54–79 years) were analyzed. Patients included in the study were patients undergoing preventive screening for glaucoma. One eye of each subject was randomly selected for anterior segment imaging. The exclusion criteria were as follows: (I) history of eye surgery or injury; (II) presence of anterior eye segment diseases (e.g., anterior uveitis, glaucoma, lens subluxation, or high myopia); (III) presence of corneal abnormalities affecting imaging (e.g., corneal opacity, corneal degeneration and dystrophies, or corneal scarring); (IV) wearing contact lenses by the participant; (V) confirmation or suspicion of the corneal ectasia; and (VI) poor fixation of the subject leading to low quality of the obtained images and lack of cooperation with the researcher. Each participant was tested for both anterior and posterior probability of corneal ectasia (ESI-A and ESI-P) using build in software of SS-OCT CASIA 2. Both confirmation and just a suspicion of corneal ectasia (any measurement > 0%) in this protocol was the criterion of exclusion from the study. Eyes with refractive error higher than or equal to ±2.0 D were also excluded. The cataract severity was measured in the dilated eye with 1% tropicamide and 10% phenylephrine hydrochloride 15 min before examination according to the Lens Opacities Classification System III (LOCS III) [17] using a slit lamp at maximum illumination without light filtering. This classification contains six degrees of the extent of nuclear opacity (NO1–NO6) and nuclear color (NC1–NC6). We only included participants with a nuclear opacity NO1 and nuclear colorNC1 in our study.

The present study was approved by the Bioethical Committee of the Medical University of Silesia in Katowice (KNW/0022/KB1/129/18), and all volunteers provided written informed consent to participate in the study. Imaging of the anterior segment of the eye was performed with CASIA2 (Tomey Corporation). All measurements were performed three times during one visit by the same experienced operator in the same sequence. During imaging, the participants were asked to fix their eyesight on a marker. The examination was performed after establishing and aligning the optical axis of the eye with the device using the following protocols: “Lens Biometry,” “AS Global Scan,” and “Corneal Map.” Parameter measurements were automatically generated by software embedded in the device. The “Lens Analysis” module in “Lens Biometry”, “3DAnalysis” module in “AS Global Scan,” and “Corneal Topography” module in “Corneal Map” were used. The acceptance criteria for the image were as follows: clearly visible scleral spur, angle, and iris. The analyzed parameters of the anterior segment were divided into three main groups, namely parameters describing the cornea, lens, and iridocorneal angle. All parameters are described in detail in Table 1.

Statistical analysis was performed in Microsoft Excel 2007 and Statistica (version 13.1.pl, StatSoft, Poland) and using the Panda and Pingou in packages in Python, StatSoft, Poland. A *p* < 0.05 was considered significant. The normality of data distribution was assessed using Shapiro–Wilk test. Significant between-group differences were analyzed using Student’s *t*-test for normally distributed data and Mann–Whitney *U*-test for non-normally distributed data. The correlations between parameters were tested using Spearman correlation coefficients. Kruskal—Wallis test was used to to explore the relationship of each parameter with age. For each parameter TRT—test-retest repeatability, CoV—within subject coefficient of variation, and ICC—intraclass correlation coefficient were calculated. 

## 3. Results

Individual parameters of the anterior segment of the eye are summarized in Table 2 and Table 3, as well as in Figure 1, Figure 2 and Figure 3. Very good intra-visit repeatability was obtained asindicated by CoV(%) (range (−0.49%)–0.33%), as well by the ICC (range 0.979–0.998). No significant differences were found for any parameter according to age groups. Mean intraocular pressure was 15.38 mmHg (SD± 3.09).

### 3.1. OCT Parameters of the Cornea

Keratometric values in the flattest and steepest meridians of the anterior and posterior corneal surfaces (Ks, Kf, pKf, and pKs) were significantly different between males and females.The parameters of the anterior surface of the cornea (Ks and Kf) were normally distributed in the study population, whereas parameters of the posterior surface of the cornea (pKs and pKf) were non-normally distributed (*p* < 0.05, Shapiro–Wilk test). Mean (±SD) Kf in females was higher (by 1.14D (±0.3)) than that in males.Similarly, pKf and pKs were significantly different between males and females (pKs, *p* < 0.001; pKf*p* < 0.01, Mann–Whitney *U*-test)with a mean difference of 0.16D (±0.04) and0.14D (±0.03), respectively.

### 3.2. Pachymetry

Mean (±SD) difference in central corneal thickness (CCT) between females and males was 12.97 (±6.24) µm, andthe differences between the two groups were near significant (*p* = 0.047, *t*-test). Moreover, CCT was significantly correlated with Ks and Kf (*p* < 0.001,) (Figure 1 and Figure 2).

### 3.3. Fourier Analysis of the Cornea

Fi-3-sph and Fi-6-Sph were significantly different between males and females (both *p* < 0.001, *t*-test); however, Fi-3-Reg, Fi-3-Ass, and Fi-3-HO were not significantly different between the two groups (*p* = 0.80; *p* = 0.53; and *p* = 0.27, respectively, Mann–Whitney *U*-test).There were significant correlations between Fi-3-sph and pKs (Figure 3),between Fi-3-Reg and anterior chamber depth (ACD), and between Fi-6-Reg and ACD (*p* < 0.05).

### 3.4. Parameters of the Iridocorneal Angle

The parameters of the iridocorneal angle were not significantly different between males and females. The iridocorneal angle was the smallest in the upper quadrant for all distances from the top of the angle (50.6° at 250 µm, 43.7° at 500 µm, and 40.6° at 750 µm). Although the iridocorneal angle was significantly smaller at the distances of 250 and 750 µm (*p* < 0.05). For all distances, the angle was the largest in the lower quadrant, but there were no significant differences between distances. Moreover, the angle opening distance (AOD) was the smallest in the upper quadrants, and at 750 µm, AOD in the upper quadrant was 25% shorter than that in the lower quadrants (0.69 vs. 0.86 mm). Moreover, the angle recess area (ARA) and trabecular iris space area (TISA) were the smallest in the upper quadrants and the highest in the lower quadrants, albeit without significant differences between the quadrants. The remaining parameters are presented in Table 3. The least useful sections were obtained for the upper quadrant.

### 3.5. Iris Parameters

All analyzed iris parameters were normally distributed (*p* < 0.05), and there were no significant differences in iris parameters between sexes. Iris thickness was significantly (*p* < 0.05) different depending on the distance from the root. Iris thickness at each angle at a distance of 2000 µm was significantly greater than that at a distance of 750 µm from the root. Iris thickness was not significantly different among the quadrants at a distance of 750 µm but significantly different at a distance of 2000 µm, where iris was significantly thicker on the quadrant side than on the other side (*p* < 0.05). In case of iris cross-sectional area, there were no significant differences among the quadrants. Convex iris dominated among all examined eyes, with an average deviation of +0.048 mm from a straight line. Concave iris was present in 44 (26.7%) of all examined eyes.

## 4. Discussion

Knowledge of the reference values of the anterior segment parameters in different age groups and sexes is seemingly essential in routine clinical practice and can allow for the correct diagnosis and monitoring of the dynamics of corneal and lens lesions and iridocorneal angle. In the present study, many of the analyzed parameters of the cornea were significantly different between sexes. Moreover, keratometric values in the flattest and steepest meridians of the anterior and posterior surfaces of the cornea, CCT, and FI3-sph in Fourier analysis were significantly different between sexes. Similar sex-specific differences have also been reported by in the literature, but on the basis of different imaging modalities. In a study using Oculus Keratograph, Sharif et al. [18] showed significant differences in the steep meridian of the corneal astigmatism keratometry between sexes; however, although the authors did not analyze the curvature of the posterior cornea [18], there were sex-specific differences is CCT. Moreover, in another study using SS-OCT, Xie et al. [18] found that CCT was 547.56 µm, which is almost identical to the value reported in the present study (546.65 µm) [19]. Furthermore, Hua et al. [20] analyzed mean CCT and certain corneal parameters using RTVue (FD-OCT); although their CCT value (544.98 ± 33.09 µm) was comparable to our value (546.65 ± 28.69) [20], they did not evaluate sex-specific differences in this parameter. The Namil study was a multicenter study including patients over 40 years of age who were residents of central South Korea; in that study, mean CCT was slightly low (530.9 µm), but this parameter was significantly different between sexes, being 5.9 µm thicker in males (*p* = 0.001) [21]. In a study by Wang et al. [22] on patients with myopia, CCT was significantly higher in males; moreover, in that study, the corneal epithelial thickness was correlated with sex in the control group, being higher in males [22]. Accurate measurement of the total power of the cornea is important for many diagnostic and therapeutic applications. The total corneal refractive power calculated using FD-OCT is characterized by high repeatability (up to 0.19 diopters), and this repeatability directly depends on the scanning speed of the device, which translates into minimizing the error resulting from image blur. In the present study, mean total corneal power for the 3 and 6 mm central circles was lower than the mean keratometric values by 0.02 D and 0.20 D, respectively, albeit not significantly different. In a similar study using RTVue, this value for the 3 mm central circle was significantly lower (by 1.21 D), which is not consistent with the results of the present study [23].The results of Fourier indices of corneal parameters obtained in the present study are partly consistent with those obtained in a study by Tanabe et al. [24], who calculated the above parameters using Fourier analysis on the basis of images obtained with videokeratography; the mean values of the four main parameters, namely spherical power, regular astigmatism, asymmetry, and higher-order irregularities, were 43.97, 0.5, 0.35, and 0.11 D, respectively, in Tanabe et al.’s study [24] and 43.09, 0.5, 0.24, and 0.15 D, respectively, in our study. Hua et al. [20] also assessed the mean power of the cornea using RTVue. For the 3 mm central circle, the power was 43.01 ± 1.53 D, which is practically identical to the value obtained in the present study (43.09 ± 1.27 D). In the present study, the filtration angle was significantly smaller in the upper quadrant. This knowledge is not new—in 1956, in his gonioscopic examination of the filtration angle, Phillips [25] showed that the filtration angle was the smallest in the upper quadrants. Notwithstanding, the use of modern technologies has enabled the documentation of such observations. In 2005, using ultrasonography, Kunimatsu et al. [26] confirmed that the filtration angle was the smallest in the upper and lower quadrants [26]. In addition, Narayanaswamy et al. [27] confirmed these observations using anterior segment (AS)-OCT (Visante OCT).The filtration angle analyzed with this generation of OCT devices was the smallest in the upper and lower quadrants. Additionally, the possibility of the coexistence of open angles in the nasal and temporal quadrants and their upward closure was highlighted [27]. The present study using the latest AS-OCT platform confirmed all these conclusions. In the present study, there were no significant differences in filtration angle between the nasal and temporal quadrants. Lack of such differences has also been reported in literature [19]. We also analyzed ACD in the present study and reported the mean value of 3.1 mm. This value is similar to that reported in the literature (2.97 [19], 2.95 [28], and 2.96 [29] mm). Moreover, ACD was correlated with sex in our analysis. Kato et al. [30] in the study among Japanese residents have reported similar associations, with the anterior chamber being shallower in females [30]. Other parameters of the iridocorneal angle in the present study were similar to those reported in literature [29]. Although mean values are available, detailed parameters for each of the four anterior chamber quadrants have not been reported.

The clinical utility of AS-OCT is limited by several factors. First, there is little evidence of the correlation between AS-OCT angle configuration and physiological measurements related to the risk of developing glaucomatous lesions (e.g., intraocular pressure). Second, at present, there is no standardized, unequivocal definition of angle closure based on OCT analysis, which can help identify patients at a higher risk of having or developing angle-closure glaucoma [31]. In the present study, we used population data of previously undiagnosed and untreated Caucasian patients to characterize the angle configuration parameters measured with AS-OCT and to determine the range of mean values for the healthy population. In previous studies, several parameters of the anterior segment OCT (e.g., ACV, ACD, AOD, TISA, and ACA) have been found to be useful to detect filtration angle closure [26,32]. To distinguish between open and closed angle of percolation, anterior chamber volume (ACV) has been used for many years. In addition, many previous studies have demonstrated the utility of ACV in differentiating between these angle configurations [33,34,35,36]. Manual measurements on one or more sections through the anterior chamber cannot accurately reflect the configuration of the entire chamber, because total ACV depends on values obtained on individual sections. In this context, the use of SS-OCT can enable rapid and accurate imaging of the anterior chamber morphology. Feili et al. [37] showed that among the parameters obtained using SS-OCT, ACV was the most effective in distinguishing the filtration angle configurations. At a cut-off of 98.1 mm^3^, the sensitivity was 90.0% and specificity was 97.1% for distinguishing the open and closed angles. In that study, for the control group with an open angle of filtration, mean ACV was 128.6 mm^3^ [37]; this value was higher in our study (154.6 mm^3^).

The conducted study is not without certain limitations. The results only refer to Caucasians and the direct comparison with other races could be characterized by certain degree of bias. A larger series is needed to further divide eyes into age categories to assess reliable cross-sectional results. Moreover, the impact of factors, such as disturbance of the tear film, internal/indoor environmental factors cannot be ruled out with the respect of the variability of the obtained results. 

In summary, thanks to the well-established utility of OCT in the diagnosis and monitoring of diseases of the anterior segment of the eye, we witnessed the dynamic and gradual progress of OCT techniques and devices as well as the development of software, all of which allow for obtaining a large number of results per unit time. Therefore, a normative database of parameters of the cornea, filtration angle, and anterior chamber using SS-OCT must be established for the correct diagnosis of diseases, such as keratoconus, corneal dystrophies, glaucoma, or pseudoexfoliation syndrome. Of note, the imaging results of the anterior segment of the eye obtained using various devices with different operating principles should not be used interchangeably.

## 5. Conclusions

SS-OCT allows for the analysis of parameters of the cornea, anterior chamber, lens, and iridocorneal angle and is thus clinically useful. Sex-specific differences in the analyzed parameters should be taken into account during the diagnosis of corneal diseases. Angle configuration is an important marker for glaucoma diagnosis and drainage implant surgery. Measurements with Casia 2 is characterized by very good repeatability.

## Figures and Tables

**Figure 1 jcm-10-01094-f001:**
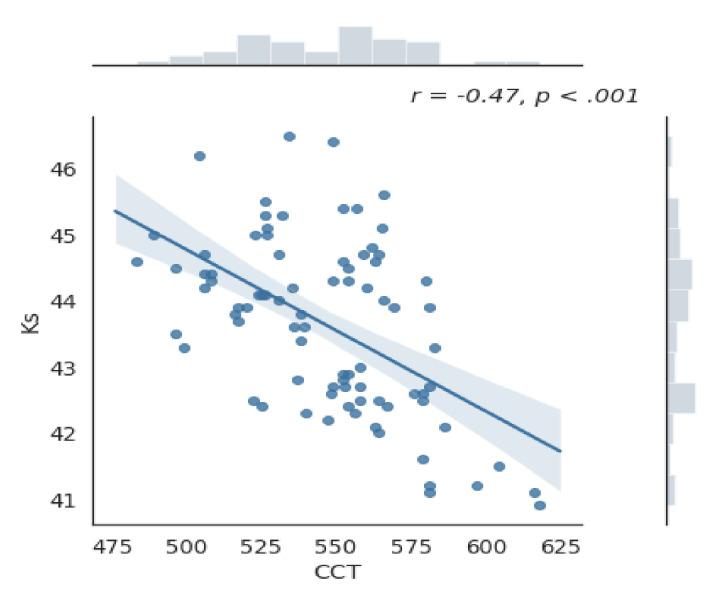
Graph of the correlation between the steep meridian of the anterior surface keratometry (Ks) and central corneal thickness (CCT). Correlation coefficient *r* = −0.47; *p* < 0.001.

**Figure 2 jcm-10-01094-f002:**
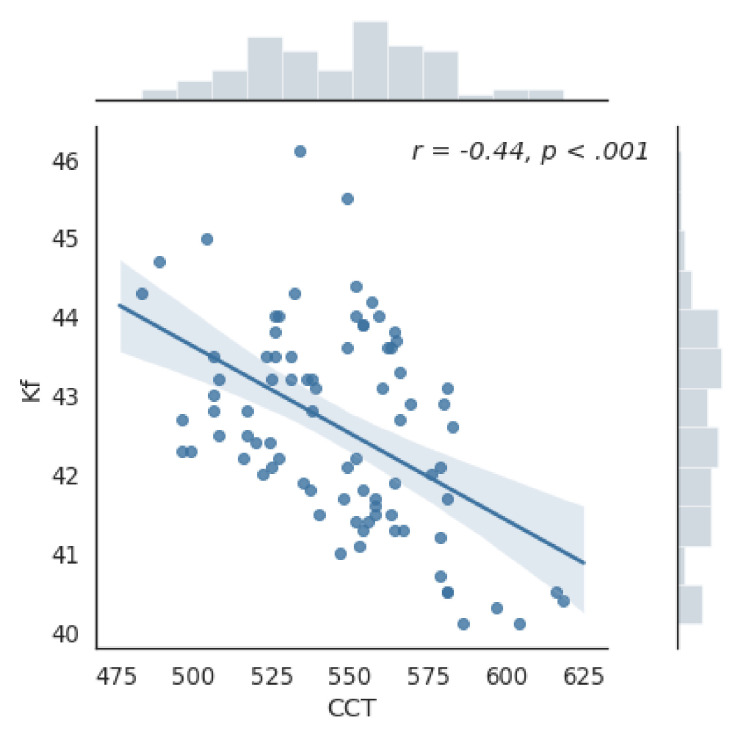
Graph of the correlation between the value of the plane meridian of anterior keratometry (Kf) and the thickness of the central cornea (CCT). Correlation coefficient *r* = −0.44; *p* < 0.001.

**Figure 3 jcm-10-01094-f003:**
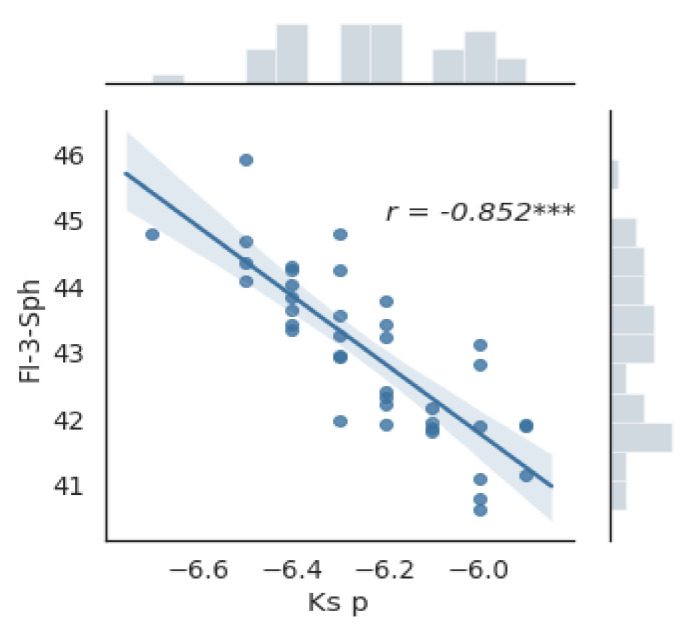
Graph of the relationship between the value of the corneal spherical refractive power component (Fi-3-sph) and the steep meridian of the keratometry of the posterior surface of the cornea. Correlation coefficient *r* = −0.852; *p* < 0.0001; *** *p* < 0.001

**Table 1 jcm-10-01094-t001:** Analyzed parameters of the anterior segment of the eye.

Parameter	Development of an English Abbreviation	Description of the Parameter	Unit
**Cornea Parameters**			
Ks	Keratometry steep	steep meridian value of keratometry	(D)—diopter
Kf	KKe Keratometry flat	flat meridian value of keratometry	(D)—diopter
pKs	Posterior keratometry steep	value of the steep meridian of the posterior curvature of the cornea	(D)—diopter
pKf	Posterior keratometry flat	value of the flat meridian of the posterior curvature of the cornea	(D)—diopter
CCT	Central corneal thickness	corneal thickness in the axis of the examination (central)	(µm)—micrometers
Fi3(6)-Sph	Fourier index spherical	the spherical refractive power component of the cornea obtained by Fourier analysis of the topographic data of the cornea with a diameter of 3, (6) mm. Corresponds to the zero-order component in Fourier analysis	(D)—diopter
Fi3(6)-Reg	Fourier index regular	the component of regular corneal refractive power astigmatism obtained by Fourier analysis of corneal topographic data with a diameter of 3, (6) mm. Corresponds to the second order component in the Fourier analysis	(D)—diopter
Fi3(6)-Asy	Fourier index assymetry	asymmetric refractive power component of the cornea obtained by Fourier analysis of corneal topographic data with a diameter of 3, (6) mm. Corresponds to the first-order component in Fourier analysis	(D)—diopter
Fi3-HO	Fourier index Higher Order	higher order irregularities—calculated by combining tertiary and higher components in Fourier analysis	(D)—diopter
**Lens parameters**			
Tilt	Tilt	the tilt of the lens axis in relation to the vision axis	(°)—degrees
Decent	Decentralization	the decentration of the lens in relation to the visual axis	(mm)—millimeter
**Filtration angle parameters**			
ACD	Anterior chamber depth	Anterior chamber depth	(mm)—millimeter
ACV	Anterior chamber volume	Anterior chamber volume	(mm^3^)—cubic millimeter
AOD 250,500,750	Angle opening distance	angle opening distance—the distance between two points located on the posterior surface of the cornea and the front surface of the iris at a distance of 250,500,750 µm from the peak of the filtration angle	(mm)—millimeter
ARA 250,500,750	Angle recess area	the recess-side area of the filtration angle delimited by a section defining AOD250,500,750	(mm^2^)—square millimeter
TISA 250,500,750	Trabecular Iris space area	the area obtained by subtracting from the ARA250 surface the area of the cavity of the angle delimited by the segment joining the spur of the sclera from its mirror point determined on the iris	(mm^2^)—square millimeter
TIA 250,500,750	Trabecular Iris angle	angle defined by 3 points—peak of the filtration angle, and points AOD250,500,750	(°)—degrees

**Table 2 jcm-10-01094-t002:** Comparison of the results of the normally distributed parameters of the anterior segment in the studied groups.

	Study Group (*n* = 166)		Women (*n* = 92)		Men (*n* = 74)		Average Difference	*p*
Parameter	Mean ± Standard Deviation	IC 95	Mean ± Standard Deviation	IC 95	Mean ± Standard Deviation	IC95		
Ks(D)	43.63±1.3	43.34–43.92	44.09 ±1.2	43.73–44.44	43.07 ±1.23	42.65–43.47	1.02	<0.0001*
Kf(D)	42.59 ± 1.27	42.32–42.87	43.1 ±1.14	42.76–43.44	41.96 ±1.16	41.57–42.34	1.14	<0.0001*
CCT (µm)	546.65 ± 28.69	540–553	540.87 ±21.52	534–547	553.84 ± 34.64	542–565	12.9	0.04 *
ACD (mm)	3.1 ± 0.31	3.03–3.17	3.04 ±0.25	2.96–3.11	3.18 ±0.36	3.06–3.3	0.14	0.04 *
ACV (mm^3^)	154.6 ± 37.42	134.4–178.3	146.7 ± 34.67	123.5–164.6	167.5 ± 41.45	140.1 – 182.8	20.3	0.11 *
FI-3-Sph (D)	43.09 ± 1.27	42.81–43.37	43.58 ±1.16	43.24–43.93	42.48 ±1.15	42.1–42.86	1.1	<0.0001 *
FI-6-Sph (D]	42.91 ± 1.25	42.64–43.19	43.40 ±1.13	43.06–43.73	42.32 ±1.44	41.93–42.7	1.08	<0.0001 *
TILT (°)	5.08 ± 1.2	4.82–5.35	4.9 ±1.02	4.59–5.2	5.31 ± 1.37	4.85–5.75	0.41	0.13 *
**Comparison of the results of the non—normally distributed parameters of the anterior segment in the studied groups**								
	Study group (*n* = 166)		Women (*n* = 92)		Men (*n* = 74)		*p*	
Parameter	Median	Interquartile range	Median	Interquartile range	Median	Interquartile range		
pKs(D)	−6.3	(−6.4)–(−6.1)	-6.3	(−6.5)–(−6.2)	−6.2	(−6.3)–(−6.0)	<0.001**	
pKf (D)	−5.9	(−6.1)–(−5.8)	-5.95	(−6.1)–(−5.9)	−5.8	(−6.0)–(−5.7)	<0.01 **	
FI-3-Reg (D)	0.48	0.33–0.65	0.485	0.34–0.63	0.47	0.28–0.69	0.8 **	
FI-3-Asy (D)	0.22	0.14–0.31	0.24	0.16–0.30	0.20	0.14–0.31	0.53 **	
FI-3-HO (D)	0.15	0.12–0.17	0.14	0.11–0.17	0.15	0.13–0.18	0.267 **	
FI-6-Reg (D)	0.47	.32–0.67	0.46	0.36–0.63	0.53	0.31–0.68	0.68 **	
FI-6-Asy (D)	0.31	0.20–0.40	0.33	0.23–0.40	0.23	0.17–0.36	0.19 **	
FI-6-HO (D)	0.16	0.14–0.19	0.16	0.13–0.20	0.16	0.15–0.18	0.48 **	
Decent (mm)	0.18	0.14–0.27	0.17	0.13–0.23	0.19	0.16–0.31	0.16 **	

*—T-student test; **—Mann U Whitney test.

**Table 3 jcm-10-01094-t003:** Comparison of the parameters of the filtration angle and the iris.

Parameter.	Mean ± Standard Deviation	IC95	Mean ± Standard Deviation	IC95	Mean ± Standard Deviation	IC95	Mean ± Standard Deviation	IC95
	0°		90°		180°		270°	
250 µm								
AOD (mm)	0.39 ± 0.19	0.34–0.43	0.38 ± 0.13	0.33–0.43	0.39 ± 0.15	0.36–0.42	0.46 ± 0.2	0.40–0.52
ARA (mm^2^)	0.1 ± 0.05	0.09–0.12	0.09 ± 0.04	0.08–0.11	0.1 ± 0.04	0.09–0.11	0.13 ± 0.07	0.11–0.15
TISA (mm^2^)	0.08 ± 0.04	0.07–0.09	0.08 ± 0.03	0.07–0.09	0.08 ± 0.03	0.07–0.09	0.10 ± 0.04	0.08–0.11
TIA (°)	52.4 ± 13.73	47.4–57.3	50.6 ± 14.37	47.5–53.6	52.1 ± 13.96	49.1–55.1	54.6 ± 14.66	50.6–58.5
500 µm								
AOD (mm)	0.57 ± 0.25	0.51–0.62	0.53 ± 0.18	0.46–0.6	0.56 ± 0.21	0.51–0.61	0.65 ± 0.27	0.58–0.73
ARA (mm^2^)	0.22 ± 0.11	0.19–0.24	0.21 ± 0.07	0.18–0.24	0.22 ± 0.09	0.2–0.24	0.27 ± 0.13	0.23–0.3
TISA (mm^2^)	0.2 ± 0.09	0.18–0.22	0.19 ± 0.06	0.17–0.22	0.2 ± 0.07	0.18–0.22	0.24 ± 0.1	0.21–0.26
TIA (°)	44.3 ± 12.1	41.6–46.8	43.7 ± 11.8	39.5–48.0	44.9 ± 11.8	42.3–47.4	47.9 ± 13.8	44.2–51.6
750 µm								
AOD (mm)	0.76 ± 0.31	0.69–0.82	0.69 ± 0.22	0.61–0.77	0.76 ± 0.27	0.7–0.82	0.86 ± 0.31	0.78–0.95
ARA (mm^2^)	0.39 ± 0.17	0.35–0.43	0.36 ± 0.12	0.32–0.41	0.39 ± 0.14	0.35–0.42	0.46 ± 0.2	0.4–0.51
TISA (mm^2^)	0.37 ± 0.16	0.33–0.4	0.35 ± 0.11	0.3–0.39	0.37 ± 0.13	0.34–0.4	0.43 ± 0.17	0.38–0.47
TIA (°)	42.15 ± 11.3	39.7–44.6	40.6 ± 10.1	37.0–44.3	42.7 ± 10.8	40.3–45.0	45.7 ± 12.1	42.5–48.9
Iris parameters								
IT 750 (mm)	0.37 ± 0.07	0.35–0.39	0.37 ± 0.08	0.34–0.4	0.36 ± 0.06	0.35–0.38	0.36 ± 0.06	0.35–0.39
IT 2000 (mm)	0.42 ± 0.07	0.4–0.44	0.49 ± 0.08	0.46–0.52	0.41 ± 0.07	0.39–0.43	0.41 ± 0.07	0.39–0.42
I-Curv(mm)	0.042 ± 0.11	0.01–0.06	0.048 ± 0.09	0.01–0.09	0.054 ± 0.1	0.03–0.07	0.046 ± 0.09	0.02–0.07
I-Area (mm^2^)	1.29 ± 0.17	1.25–1.32	1.39 ± 0.15	1.32–1.46	1.25 ± 0.17	1.21–1.29	1.26 ± 0.16	1.22–1.29

## Data Availability

The data presented in this study are available in Chair and Clinical Department of Ophthalmology, Faculty of Medical Sciences in Zabrze, Medical University of Silesia in Katowice, 40-055 Katowice, Poland.

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
