# Peer review of "Swept Source Optical Coherence Tomography Analysis of the Selected Eye’s Anterior Segment Parameters"

_jcm, 2021, doi:10.3390/jcm10051094_

Round 1

Reviewer 1 Report

You did not show the data of the cataract in these participants who are

relatively older. Cataract influences the depth of anterior chamber.

Are there any contact lens users or keratoconus patients?

I think the number of 166 volunteers is not enough to conclude your result. Please calculate the power of the number for the study.

Please provide refraction of the volunteers, because it is really important to discuss the depth of anterior chamber.

Intra ocular pressure is also important factor to study the corneal thickness.

Reviewer 2 Report

I read with interest the manuscript entitled "Swept source optical coherence tomography analysis of the selected eye’s anterior segment parameters."

The overal manuscript is well written and organized.

However, the authors should address the following issues:

The authors state that the measurements were repeated three short times, thus I expected an evaluaiton of intra-visit repeatability;

the patients are relatively older, the authors should specify the reasons, e.g., if they have been recruited among eye services for AMD or cataract. This issue should be addressed a limit.

The authors stated that some parameters are non normally distributed. For these parameters, median and interquartile range would be reported.  

The authors stated that the small sample did not enable to evaluate the parameters according to age categories. I suggest to explore the relantionship of each parameter with age, considering  

Since several comparisons have been done and reported, the authors should consider to correct for multiple comparison.

Round 2

Reviewer 1 Report

Please provide the words of :Women (n=92) Men (n=74) in abstract, too.
